# Effects of Heat-Moisture Treatment Whole Tartary Buckwheat Flour on Processing Characteristics, Organoleptic Quality, and Flavor of Noodles

**DOI:** 10.3390/foods11233822

**Published:** 2022-11-27

**Authors:** Yunlong Li, Wenwen Chen, Hongmei Li, Jilin Dong, Ruiling Shen

**Affiliations:** 1Institute of Functional Food of Shanxi, Shanxi Agricultural University, Taiyuan 030031, China; 2College of Food and Bioengineering, Zhengzhou University of Light Industry, Zhengzhou 450002, China; 3Henan Key Laboratory of Cold Chain Food Quality and Safety Control, Zhengzhou University of Light Industry, Zhengzhou 450002, China; 4Collaborative Innovation Center of Food Production and Safety, Zhengzhou University of Light Industry, Zhengzhou 450002, China

**Keywords:** noodles, heat-moisture treatment whole tartary buckwheat flour, processing characteristics, organoleptic quality, flavor, glycemic index

## Abstract

The effects of heat-moisture treatment whole tartary buckwheat flour (HTBF) with different contents on the pasting properties and hydration characteristics of tartary buckwheat noodle mix flour (TBMF), dough moisture distribution, cooking properties, texture properties, and flavor of noodles were studied. The results showed that the optimal additional amount of HTBF is determined to be 40%. The peak viscosity, trough viscosity, breakdown value, and final viscosity decreased significantly, and the optimal cooking time of the noodles decreased with increasing HTBF. Compared with the sample without HTBF, HTBF addition increased the water absorption of the sample and decreased its water solubility. When the amount of HTBF >30%, the content of strongly bound water in dough increased significantly; at HTBF >40%, the water absorption and cooking loss of noodles increased rapidly, and the hardness of noodles was decreased; and with HMBF addition at 60%, the chewiness, resilience, and elasticity decreased. Moreover, HMBF addition reduced the relative content of volatile alkanes, while increasing the amount of volatile alcohols. HTBF addition also elevated the content of slow-digesting starch (SDS) and resistant starch (RS) in noodles, providing noodles with better health benefits in preventing chronic diseases. These results proved the possibility of applying heat-moisture treatment grains to noodles, and they provide a theoretical basis for the research and development of staple foods with a hypoglycemic index.

## 1. Introduction

Globally, influenced by lifestyle and dietary structure, various chronic diseases are common in the population, and public awareness of the threat of chronic diseases to human health has been increasing. Thus, an increasing number of people are focusing on the nutrition and functionality of food, and low-fat and low-sugar diets are becoming increasingly popular. Tartary buckwheat (*Fagopyrum tataricum* (L.) *Gaertn*), in the genus *Fagopyrum*, *Polygonaceae* family, is one of the main buckwheat species cultivated worldwide, as well as a type of homology between medicine and food [1]. The starch content of tartary buckwheat is approximately 60–70%, which has the property of slow digestion. To a certain extent, it maintains the stability of postprandial blood sugar levels and reduces insulin secretion. Thus, it is suitable for treating diabetes and obesity. The contents of proteins in the flour of different tartary buckwheat varieties range from 9% to 15%, and they contain eight kinds of amino acids required by the human body, which the body easily absorbs [2]. The protein isolates and derived peptides in tartary buckwheat have cholesterol-lowering, antioxidant, hypoglycemic, antibacterial, and intestinal microbiota-regulating effects [3,4]. Tartary buckwheat is also rich in bioactive substances, such as flavonols, flavones, isoflavones, flavonone, flavon-3-ols, anthocyanins, flavonolignans, phenolic acids, phenolic acids, steroids, triterpenoids, and so on [5]. These are the main reasons why tartary buckwheat is considered to have a high nutritional value and to promote health. In recent years, numerous tartary buckwheat products have appeared in the market, such as tartary buckwheat flour products, tartary buckwheat baked goods, and tartary buckwheat drinks, which meet the demand for tartary buckwheat foods to a certain extent [6]. Tartary buckwheat is a type of false grain without gluten—which limits the amount of tartary buckwheat added to staple food as raw materials, such as noodles and steamed bread—and its efficacy cannot be fully utilized. Therefore, it is very important to add quality improvers or treat tartary buckwheat flour to improve the added amount and functionality of tartary buckwheat products. Heat-moisture treatment (HMT) is a physical modification method carried out under relatively low moisture content (<35%) and high temperature (80–140 °C). The process can destroy the crystal structure of starch and dissociate the double helix structure, promote the interaction of polymer chains, rearrange the broken crystals, and affect the swelling, solubility, gelatinization characteristics, particle morphology, and sensitivity to enzymatic or acidic hydrolysis of whole grain particles, thus affecting starch digestibility [7]. In practical applications, according to the target food needs, the moisture content, treatment temperature, and treatment time can be varied to achieve a better application effect [8]. HMT could make the starch contain lower rapid-digesting starch (RDS) content and higher slow-digesting starch (SDS) and resistant starch (RS) content, thus reducing the in vitro hydrolysis rate of starch [9]. The decreases in rapidly digested starch and total hydrolysis content using HMT are greater than those obtained using annealing [7]. Thus, HMT starch has great health benefits in preventing chronic diseases, especially diabetes. Noodles are a staple food in many countries, including China, Italy, and Japan. Tartary buckwheat starch gel after hydrothermal treatment has lower hardness and swelling power, making it ideal for tartary buckwheat foods such as noodles, soup, and dumplings [8]. Traditional wheat noodles have a higher sugar index and lower nutritional content than coarse-grained noodles. Madalina et al. [10] added HMT wheat flour and grape seeds to noodles to increase the nutritional value of wheat noodles. Kömürcü et al. [11] found that when HMT germinated wheat flour was added to noodles, cooking loss decreased and the chewiness and nutritional value of the germinated wheat flour noodles increased. Similarly, HMT amaranth starch noodles form a sticky and smooth surface structure with better taste and aroma [12]. However, the application of HMT flour in various grain noodles is still lacking; thus, it is very important to study the effect of HMT flour on the quality of various grain noodles.

Therefore, our research aimed to (1) develop tartary buckwheat noodles (TBN) with whole tartary buckwheat flour (TBF) and heat-moisture treatment of whole tartary buckwheat flour (HTBF), (2) characterize the effects of HTBF and different amounts of HTBF on the characteristics of noodle flour and dough, and (3) study the effects of HTBF addition on the sensory, flavor, and digestive characteristics of tartary buckwheat noodles. Data from this research will promote HTBF application in noodles and the research and development of functional cereal noodles.

## 2. Materials and Methods

### 2.1. Raw Materials

TBF produced in 2020 was purchased from Shanxi Qinggao Food Co., Ltd. (Datong, China). Wheat gluten was purchased from Fengqiu Huafeng Powder Industry Co., Ltd. (Xinxiang, China). Composite gel polysaccharides and transglutaminase were purchased from Henan Wanbang Industry Co., Ltd. (Ningxia, China).

### 2.2. Preparation of Tartary Buckwheat Noodle Mix Flour (TBMF)

The TBF moisture content was adjusted to 30%, the TBF was sealed in a bag at 4 °C for 24 h to balance the moisture. Subsequently, TBF was transferred to a steel pan, heated in an oven at 110 °C for 90 min, and then dried at 45 °C and milled through a 100-mesh sieve. 

Seven types of wholegrain noodle samples were produced by heat-moisture treatment HTBF substituting 0%, 10%, 20%, 30%, 40%, 50%, and 60% of native whole tartary buckwheat flour. The number of quality improvers in each sample was consistent, including 14.7% protein powder, 1.50% complex gel polysaccharide, and 0.1% TG enzyme.

### 2.3. Processing Characteristics of TBMF

#### 2.3.1. Pasting Property

The pasting properties of the samples were analyzed using a Rapid Visco Analyzer (RVA4500, Perten Instruments, Warriewood, NSW, Australia), as described by Zhang et al. [13] with some modifications. Samples (3 g, dry weight basis) were accurately weighed into RVA canisters, followed by distilled water (25 mL). The suspensions were equilibrated in the RVA at 50 °C for 1 min then heated to 95 °C at a rate of 6 °C/min. The samples were subsequently maintained at 95 °C for 5 min, cooled to 50 °C at a rate of 6 °C/min then held at 50 °C for 2 min. The heating process was accompanied by a constant shear at 960 rpm for the first 10 s to disperse the sample, and then the speed was reduced to 160 rpm for the rest of the experiment. All measurements were performed in triplicates.

#### 2.3.2. Water Solubility Index (WSI) and Swelling Power (SP)

WSI and SP were analyzed at 25 °C, 37 °C, and 50 °C [14]. The suspension was prepared separately with 1 g of TBMF in 20 mL of solution. The suspensions were stirred for 30 min at different temperatures and centrifugation (2683× *g* for 20 min). The supernatants were then oven-dried (105 °C) until the weight was constant. All analyses were performed in triplicates. The water absorption index (WAI; Equation (1)) and the water solubility index (WSI, Equation (2) was calculated as follows:(1)$$\mathrm{WAI}=\frac{\mathrm{Wet}\;\mathrm{sediment}\;\mathrm{weight}}{\mathrm{Dry}\;\mathrm{sample}\;\mathrm{weight}}\times 100$$
(2)$$\mathrm{WSI}\left(\%\right)=\frac{\mathrm{Dry}\;\mathrm{supernatant}\;\mathrm{weight}}{\mathrm{Dry}\;\mathrm{sample}\;\mathrm{weight}}\times 100$$

#### 2.3.3. Water Distribution of Tartary Buckwheat Dough

The water distribution in the dough was measured using low-field nuclear magnetic resonance (LF-NMR). The transverse relaxation time (T2) was measured using the Carr-Purcell-Meiboom-Gill method [15]. Dough (1.0 g) was placed in an NMR tube and placed in a probe with a low-resolution (18 MHz) pulsed NMR spectrometer (NMI20, Shanghai Electronic Technology Co., Ltd., Shanghai, China) for Carr-Purcell-Meiboom-Gill pulse sequence testing. The number of sampling points was 166,398, the echo time was 0.30 ms, the number of echoes was 1000, and 4 scan reiterations were acquired.

### 2.4. The Organoleptic Quality of TBN

#### 2.4.1. Cooking Characteristics

The cooking loss of TBN was determined under the optimal cooking time. The optimal cooking time was recorded as TBN cooking until the core was cooked. Briefly, 40 g TBN was recorded and weighed accurately, placed in 500 mL boiling water, and cooked in a slightly boiling state until the optimal cooking time was reached. Then, the TBN was removed and the weight of the complete TBN was recorded (x_1_). The soup was placed in a constant 250 mL beaker (m_0_) at room temperature and baked in a 105 °C oven until it reached constant weight (m_2_). The cooking loss (CL, Equation (3)) and the TBN breakage rate were calculated as follows [16]:(3)$$\mathrm{Cooking}\;\mathrm{loss}=\frac{{(m}_{2}{-m}_{0})}{\left[m_{1}\times (1-w)\right]}\times 100\%$$
where w is the moisture content of TBN (%).

#### 2.4.2. Texture Characteristics of Cooked TBN

After cooking for the optimal time, the TBN was quickly removed and soaked in cold water for 10 s, then removed and covered with plastic wrap [17]. Three noodles were laid on the test platform in parallel, and a P36/R probe was used for the TPA test. The speed before the test was 1.0 mm/s, during the test was 0.2 mm/s, and after the test was 1.0 mm/s. The triggering force was 5 g, the compression deformation was 75%, and the interval between the two compressions was 3 s.

#### 2.4.3. Volatile Components

The volatile TBN components were measured using headspace solid-phase microextraction (HS-SPME) coupled to a GC (7890 B, Agilent Technologies, Santa Clara, CA, USA) with an MS (5977A, Agilent Technologies; Santa Clara, CA, USA) detector and MSD Chemstation for data processing [18]. The GC was equipped with RTX-5 MS elastic quartz capillary column (30 m × 0.25 mm × 0.25 um). Briefly, 5.00 g TBN was weighed in a 20 mL sample bottle, sealed with a Teflon-lined septum and screw cap, and balanced in a preset 80 °C constant temperature water bath for 10 min. A 50/30 μm DVB/CARΒOXEN™/PDMS fiber was exposed to the vial to extract VOCs emitted from TBN for 40 min. Volatile compounds were identified by matching their total ion (*m*/*z* from 50 to 300) mass spectra with the database NIST 11. 

### 2.5. Digestibility In Vitro

The method described by Yu et al. was used to determine the extent of in vitro starch hydrolysis [19]. In total, 6 mL deionized water and samples (containing 100 mg starch, dry weight basis) were mixed, and the solution was mixed with 2 mL pepsin solution (37 °C shaking for 30 min). The pH of suspensions was modified to 7, and 3 mL amylase-amyloglucosidase solution was added to the above solution and shaken for 180 min at 37 °C. At special time intervals (0, 20, 40, 60, 80, 100, 120, 140, 160, and 180 min), to inactivate the enzymes, 1 mL of aliquots and 9 mL of 0.3 M Na_2_CO_3_ solution were added. After centrifugation, a 3,5-dinitrosali-cyclic acid assay was used to determine the reducing sugar content of the supernatant. The RDS content (RDS, Equation (4)), SDS content (SDS, Equation (5)), and RS content (RS, Equation (6)) are calculated as follows:(4)$$\mathrm{RDS}\;(\%)=\frac{{(G}_{20}-\mathrm{FG})}{\mathrm{TS}}\times 100$$
(5)$$\mathrm{SDS}\;(\%)=\frac{{(G}_{120}-G_{20})\times 0.9}{\mathrm{TS}}\times 100$$
(6)$$\mathrm{RS}\;(\%)=\frac{\mathrm{TS}-\mathrm{RDS}-\mathrm{SDS}}{\mathrm{TS}}\times 100$$
where FG is the free reducing sugar content in the flour before enzymatic hydrolysis (mg), TS is the total starch content (mg), G_20_ is the reducing sugar content (mg) produced after 20 min of enzymatic hydrolysis, and G_120_ is the reducing sugar content (mg) produced after 120 min of enzymatic hydrolysis.

### 2.6. Statistical Analysis

All experiments were conducted in triplicate, and the results are expressed as mean ± standard deviation. The data of the various indicators were analyzed using Origin 2018 (Origin Lab, Northampton, MA, USA) and SPSS software (version 23.0; IBM Corp., Armonk, NY, USA). Analysis of variance (ANOVA) was used to determine significant differences between the results, and Tukey’s post hoc test was used to compare the means with a significance level of 0.05.

## 3. Results and Discussion

### 3.1. Pasting Property

Starch viscosity is an important factor that determines its applicability in food processing. The gelatinization characteristics of starch can affect the texture and appearance of noodles. The peak viscosity and breakdown value of noodles are positively correlated with the smoothness, taste, viscosity, appearance, and elasticity of noodles, which could reflect starch stability at high temperatures, and the final viscosity is related to the shear resistance of starch [20,21]. The pasting parameters of the noodle flour are summarized in Table 1. The addition of HTBF increased the pasting temperature of TBN, indicating that TBF thermal stability was enhanced, which was related to the interaction between amylose chains, as well as between amylose and lipids, which reduced mobility in amorphous domains [8]. The decreases in peak viscosity, trough viscosity, breakdown value, and final viscosity were significant with increasing HTBF. HMT could induce higher amylose levels in cereal starch and disrupt its chain length distribution. TBF was partially gelatinized during HMT, which is a possible reason for the decreased pasting viscosities of TBN with an increase in the substitution amount of HTBF. The decrease in trough viscosity makes noodles more susceptible to cooking erosion [22]. The setback value was used to indicate the degree of aging caused by the re-aggregation of starch particles after gelatinization. The setback value decreased with increasing HTBF. This is because the starch molecules in HTBF contain highly ordered double helix amylopectin clusters and starch lipid complexes, which reduce amylose leaching and starch viscosity [7]. In general, the addition of HTBF deteriorates noodle quality, and the deterioration phenomenon increases with an increase in the amount added.

### 3.2. Hydration Characteristics

The effect of HTBF addition on noodle processing characteristics was evaluated by measuring the changes in the water absorption index (WAI) and water solubility index (WSI) of the samples at different temperatures. As shown in Figure 1, the addition of HTBF significantly affected the WAI and WSI of the noodle flour. Compared to the sample without HTBF, the addition of HTBF increased the water absorption of the sample and decreased its water solubility. With the increase in HTBF addition, the water absorption of the sample increased slowly at first and then increased rapidly, while its water solubility slowly decreased at first and then rapidly decreased. The primary reason for the increase in water absorption was starch gelatinization during the heat-moisture treatment. Keppler [23] and Ma [24] have similar findings, and they believe that these results may be attributed to the interaction between water and carbohydrate polar groups and other polar groups in the flour. Starch gelatinization led to the dissolution of soluble substances and the leaching of amylose in flour. However, hydrothermal treatment induces amylose lipid complexes to hinder starch swelling, thereby reducing the leaching and solubility of amylose [25]. Additionally, with an increase in temperature, the water solubility and water absorption of all noodle powders increased, which could be due to the generation of more dissociated hot-water-soluble short chains. An increase in water absorption increases the elasticity and smoothness of the noodles. However, when the gluten content in the noodles is low or the gluten network strength is insufficient, the high water absorption makes the noodles soft and lacking in texture, and it increases the cooking loss and breaking rate. From the experimental results, we believe that when the amount of HTBF added does not exceed 40%, the sensory quality of tartary buckwheat noodles will not be reduced owing to excessive water absorption.

### 3.3. Water Distribution

The LF-NMR spin-spin relaxation time (T2) or transverse relaxation time constant was used to indicate the state of the water in doughs and to predict the adsorption and water-holding capacity of the dough [26]. As shown in Table 2, for all doughs, three distinct water populations were observed, centered at around 0.38–0.54 ms (T21), 9.33–17.52 ms (T22), and 114.98–285.50 ms (T23). As previously described, relaxation times of <1 ms were considered to correspond to the bound water state, and T21 represented tightly bound hydrogen protons in the gluten matrix, including the CH of amino acids and strongly bound water molecules [27]. T22 was the main peak in the CPMG sequence and was composed of exchangeable protons, reflecting the immobilized water. T23 corresponds to free water, and its motility is not bound by gluten proteins [28]. The ratio of the peak areas to the respective total areas of A21, A22, and A23 reflects the different forms of water. In all doughs, A22 was larger than A21, and A23 content was the lowest, which indicated that most of the water was attached to the surface of the starch or gluten protein, and the binding force between water and dough components was weak. With a continuous increase in HTBF addition, T21 slightly increased, and T22 and T23 significantly decreased. When the amount of HTBF was over 30%, A21 increased significantly and A22 decreased, indicating that the gluten and water were more closely combined. With the continuous increase of HTBF addition, A21 and A22 did not increase significantly, but A23 increased. These phenomena indicate that the addition of an appropriate amount of HTBF could increase the formation of rigid protons in the dough.

### 3.4. Cooking Characteristics

Cooking loss and optimum cooking time are important characteristics that determine the cooking quality of the noodles. The cooking properties of noodles are shown in Figure 2. As shown, when HTBF content was under 20%, the cooking loss did not change significantly. Subsequently, with an increase in HTBF addition, cooking loss gradually increased. When the HTBF content was over 40%, the cooking loss increased rapidly. The addition of HTBF reduced the optimal cooking time of the noodles, and the optimal cooking time decreased with an increase in HTBF addition. After HMT, the starch gel deteriorated and the dietary fiber content in HTBF increased, which interrupted the gluten network and reduced the stability of the dough. This led to the easy leaching of starch and faster water penetration in the cooking process of noodles, thus increasing the cooking loss and shortening the optimal cooking time of noodles. Kömürcü et al. [11] reported similar findings when studying the effects of germinated and heat-moisture-treated ancient wheat on some quality attributes and bioactive components of noodles.

### 3.5. TPA Test 

Textural characteristics can directly reflect the eating quality of noodles. As shown in Table 3, the addition of HTBF reduced the hardness, cohesiveness, and gumminess of the noodles and increased the adhesiveness and springiness. When the amount of HTBF added was greater than or equal to 30%, the hardness decreased significantly. Previous studies have demonstrated that changes in hardness after HMT are positively correlated with an increase in the amylose content and water-binding capacity of starch [12]. The reduction in cohesiveness and gumminess may be due to gluten formation inhibition by the addition of HTBF, which reduces dough strength. Chewiness and resilience first increased and then decreased with increasing HTBF addition. Springiness, chewiness, and resilience first showed an increasing trend with the increase in the amount of HTBF added; chewiness and resilience decreased when the amount of HTBF added reached 60%, and springiness decreased when the amount of HTBF added reached 50%. These results indicate that the overall texture of noodles is affected by HTBF. When the HTBF content was under 40%, the hardness of noodles was less affected, and the palatability of noodles could be improved to some extent. However, excessive HTBF deteriorates noodle quality.

### 3.6. Aroma Components

According to the SPME-GCMS results in Figure 3, the volatile substances of all noodle products were composed of aldehydes, alkanes, esters, alkenes, furan, ketones, benzene, phenols, sulfides, alcohols, and other classes, among which the relative content of aldehydes and alkanes was the highest. Among all the volatile substances, the most volatile alkanes were detected, and the contents of tridecane, dodecane, tetradecane, and hexadecane were high. Previous studies have shown that the flavor of cereals increases with an increase in carbon chain length and in the proportion of unsaturated hydrocarbons [29]. Hydrocarbons mainly come from the homolysis of fatty acid alkoxy radicals, but because their threshold values are generally high, it can be considered that they contribute little to buckwheat flavor [30]. Ketones are one of the main groups of carbonyl compounds, and volatile ketones are likely to be products of lipid or amino acid degradation. There are relatively few ketone compounds in buckwheat, and the ketone threshold is higher than that of aldehydes; therefore, it is not easy to perceive and has little impact on flavor characteristics [31]. The addition of HTBF decreased the relative content of volatile alkanes in the noodles and increased the amount of volatile alcohols. In noodles, the volatile aldehydes detected include nonanal, (E)-2-Nonenal, hexanal, decanal, benzaldehyde, heptanal, and benzeneacetaldehyde. Among all aldehydes, the content of nonanal(>50%) was the highest, followed by (E)-2-Nonnal, hexanal, benzaldehyde, decanal, and heptanal. Aldehydes have a lower threshold value and have a greater impact on flavor. Nonal has the flavor of oil and sweet orange, n-hexanal has the flavor of grass and fruit at low concentrations, decanal gives buckwheat hull a fruit flavor, and benzaldehyde has the sweet smell of almond; benzaldehyde and decanal mainly exist in buckwheat hull [32,33]. Therefore, aldehydes may be the main flavoring substances in tartary buckwheat noodles.

### 3.7. Digestibility In Vitro

As shown in Figure 4, the HTBF adjunction reduced the RDS content and increased the SDS and RS content in noodles. Among the starch nutrients, SDS can be digested in the small intestine five times longer than RDS, resulting in lower postprandial blood glucose. Resistant starch cannot be degraded by human gastrointestinal enzymes. After RS reaches the colon, it can be used in microbial metabolism to produce bioactive short-chain fatty acids to prevent cardiovascular disease. In general, the enzyme sensitivity of starch is affected by its source, particle size, amylose content, and crystal structure [34]. The crystallinity of HMT-modified buckwheat flour is higher than that of starch, which results in the limited accessibility of enzymes to starch chains, and the amylose lipid complex formed by the mutual or internal reaction between starch and protein or lipid leads to an increase in SDS [8,9]. With the growth of HTBF content, the SDS and RS content increased gradually with the rise of HTBF. This shows that noodles made from HTBF have potential health benefits in the prevention of chronic diseases.

## 4. Conclusions

In conclusion, the addition of HTBF increased the pasting temperature and water absorption of TBMF, decreased the viscosity and water solubility of the dough, and increased the content of strongly bound water in the dough. It also reduced the relative content of volatile alkane substances, while increasing the relative content of volatile alcohol substances in noodles. Moreover, it increased the content of SDS and RS in noodles, providing noodles with better health benefits in preventing chronic diseases. When the amount of HTBF was in the range of 0–40%, springiness, chewiness, and resilience of noodles were high, the hardness and cooking loss were moderate, and the cooking time was reduced. However, when HTBF was greater than 40%, water absorption and cooking loss of noodles increased rapidly, and hardness and cooking time were too low, resulting in the poor palatability of noodles. Through comprehensive consideration of parameters, it was recommended that HTBF at a 40% ratio would be suitable for use in noodles. The findings of this study established the application value of HTBF in noodles and provided a model for the application of heat moisture treatment grains in noodles and the development of staple foods with a hypoglycemic index.

## Figures and Tables

**Figure 1 foods-11-03822-f001:**
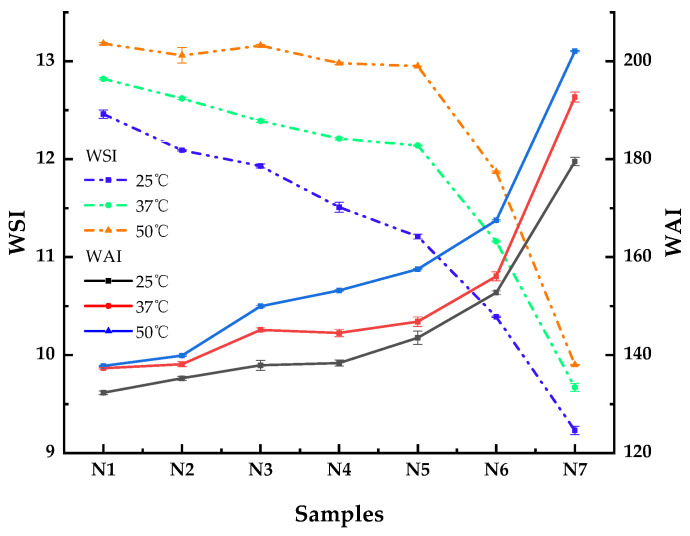
The WAI and WSI of tartary buckwheat noodle flour. N1–N7 respectively represent noodles with 0%, 10%, 20%, 30%, 40%, 50% and 60% HTBF.

**Figure 2 foods-11-03822-f002:**
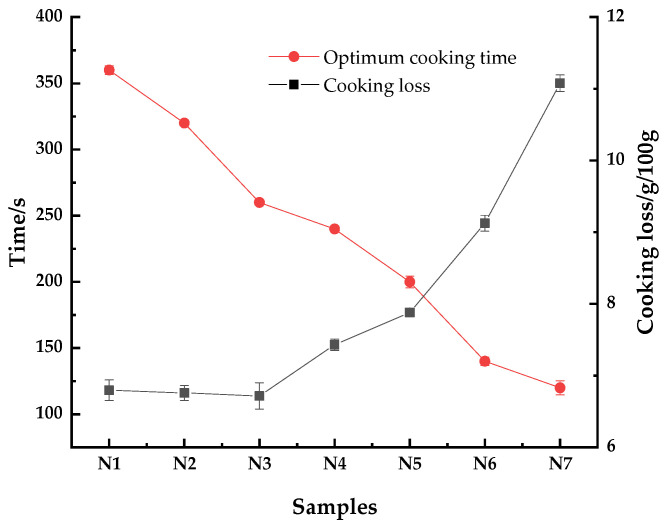
Cooking properties of the tartary buckwheat noodles. N1–N7 respectively represent noodles with 0%, 10%, 20%, 30%, 40%, 50%, and 60% HTBF.

**Figure 3 foods-11-03822-f003:**
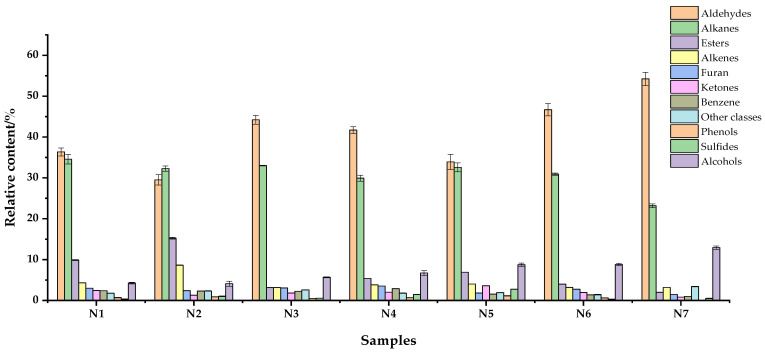
The relative content of volatile compounds in tartary buckwheat noodles. N1–N7 respectively represent noodles with 0%, 10%, 20%, 30%, 40%, 50%, and 60% HTBF.

**Figure 4 foods-11-03822-f004:**
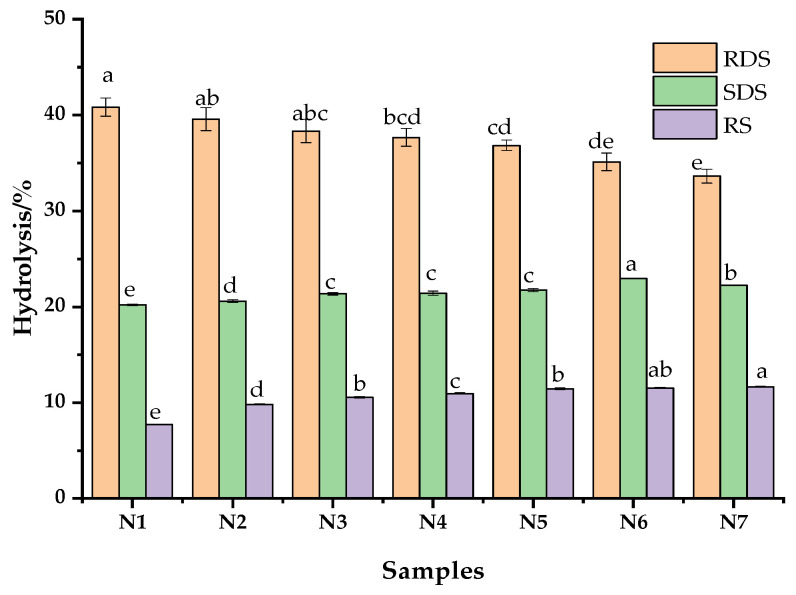
The RDS, SDS, and RS levels in tartary buckwheat noodles with different HTBF levels. Bars bearing the same letter within the same starch cultivar are not significantly different (*p* < 0.05). N1–N7 respectively represent noodles with 0%, 10%, 20%, 30%, 40%, 50%, and 60% HTBF.

**Table 1 foods-11-03822-t001:** Pasting parameters of noodle flour.

Samples	Pasting Temp (℃)	Peak Time (min)	Peak Viscosity (cP)	Trough Viscosity (cP)	Breakdown (cP)	Final Viscosity (cP)	Setback (cP)
N1	79.90 ± 1.33 ^e^	5.87 ± 0.12 ^c^	2290.00 ± 13.00 ^a^	1859.00 ± 7.33 ^a^	431.00 ± 3.67 ^a^	2997.00 ± 14.67 ^a^	1138.00 ± 21.33 ^a^
N2	83.90 ± 1.00 ^d^	5.67 ± 0.23 ^c^	1593.33 ± 15.00 ^b^	1593.00 ± 10.67 ^b^	250.33 ± 6.00 ^b^	2604.00 ± 11.33 ^b^	1011.00 ± 6.67 ^b^
N3	85.50 ± 0.67 ^cd^	5.80 ± 0.00 ^c^	1255.00 ± 11.67 ^c^	1255.00 ± 18.33 ^c^	81.00 ± 4.00 ^c^	1920.00 ± 13.33 ^c^	665.33 ± 9.67 ^c^
N4	87.90 ± 0.67 ^bc^	6.27 ± 0.00 ^b^	1020.33 ± 19.67 ^d^	1020.67 ± 6.67 ^d^	28.00 ± 4.00 ^d^	1499.67 ± 13.33 ^d^	479.00 ± 11.00 ^d^
N5	89.70 ± 1.33 ^b^	6.93 ± 0.09 ^a^	667.33 ± 13.33 ^e^	667.00 ± 13.00 ^e^	17.67 ± 4.67 ^de^	1019.00 ± 10.00 ^e^	352.00 ± 6.00 ^e^
N6	92.94 ± 1.67 ^a^	7.00 ± 0.00 ^a^	476.00 ± 15.67 ^f^	476.00 ± 32.00 ^f^	22.00 ± 5.00 ^de^	730.00 ± 16.67 ^f^	254.33 ± 3.33 ^f^
N7	--	6.87 ± 0.18 ^a^	285.00 ± 14.00 ^g^	285.00 ± 11.33 ^g^	14.00 ± 3.33 ^f^	450 ± 16.00 ^g^	165.00 ± 22.00 ^g^

Data represent the mean of three independent experiments ± standard deviation (*n* = 3). Values with different superscript letters in the same column (lowercase letters) differ significantly (*p* < 0.05). N1–N7 respectively represent noodles with 0%, 10%, 20%, 30%, 40%, 50%, and 60% HTBF. -- Represents no data detected.

**Table 2 foods-11-03822-t002:** T2 and peak areas of doughs.

Samples	T2/ms			Peak Eares/%		
	T21	T22	T23	A21	A22	A23
N1	0.40 ± 0.04 ^b^	17.52 ± 1.73 ^a^	285.50 ± 28.13 ^a^	6.27 ± 1.43 ^b^	92.62 ± 1.46 ^a^	1.11 ± 0.03 ^d^
N2	0.38 ± 0.00 ^b^	17.52 ± 1.73 ^a^	285.50 ± 28.13 ^a^	5.72 ± 0.00 ^b^	93.15 ± 0.14 ^a^	1.13 ± 0.13 ^d^
N3	0.50 ± 0.00 ^ab^	14.18 ± 0.00 ^ab^	200.92 ± 0.00 ^b^	5.84 ± 0.06 ^b^	92.73 ± 0.04 ^a^	1.43 ± 0.02 ^c^
N4	0.50 ± 0.00 ^ab^	14.18 ± 0.00 ^ab^	200.92 ± 0.00 ^b^	6.52 ± 0.91 ^b^	92.04 ± 0.94 ^a^	1.44 ± 0.03 ^bc^
N5	0.54 ± 0.05 ^a^	12.33 ± 0.00 ^bc^	163.37 ± 16.10 ^bc^	13.20 ± 0.66 ^a^	85.09 ± 0.57 ^b^	1.71 ± 0.09 ^b^
N6	0.54 ± 0.05 ^a^	12.33 ± 0.00 ^bc^	151.99 ± 0.00 ^bc^	12.65 ± 0.40 ^a^	85.65 ± 0.37 ^b^	1.70 ± 0.03 ^bc^
N7	0.50 ± 0.00 ^ab^	9.33 ± 0.00 ^c^	114.98 ± 0.00 ^c^	13.83 ± 0.69 ^a^	84.05 ± 0.63 ^b^	2.13 ± 0.06 ^a^

Data represent the mean of three independent experiments ± standard deviation (*n* = 3). Values with different superscript letters in the same column (lowercase letters) differ significantly (*p* < 0.05). N1–N7 respectively represent noodles with 0%, 10%, 20%, 30%, 40%, 50%, and 60% HTBF.

**Table 3 foods-11-03822-t003:** Texture index of tartary buckwheat noodles.

Samples	Hardness/g	Adhesiveness/g·s	Springiness	Cohesiveness	Gumminess	Chewiness	Resilience
N1	12,732.68 ± 216.64 ^a^	−77.67 ± 0.28 ^a^	0.59 ± 0.01 ^d^	0.70 ± 0.00 ^a^	7691.58 ± 59.64 ^a^	4345.17 ± 90.22 ^a^	0.35 ± 0.00 ^c^
N2	12,801.97 ± 331.25 ^a^	−85.65 ± 5.08 ^ab^	0.64 ± 0.01 ^c^	0.67 ± 0.05 ^ab^	7239.52 ± 34.87 ^b^	4404.56 ± 99.36 ^a^	0.36 ± 0.00 ^bc^
N3	12,630.35 ± 438.87 ^a^	−96.68 ± 1.69 ^b^	0.67 ± 0.01 ^c^	0.60 ± 0.05 ^bc^	7096.74 ± 4.64 ^b^	4483.78 ± 65.93 ^a^	0.37 ± 0.00 ^b^
N4	12,355.05 ± 556.80 ^ab^	−94.60 ± 2.01 ^b^	0.71 ± 0.01 ^b^	0.56 ± 0.03 ^cd^	6524.38 ± 82.97 ^c^	4602.61 ± 63.05 ^a^	0.37 ± 0.01 ^b^
N5	11,398.00 ± 249.78 ^b^	−93.46 ± 3.79 ^b^	0.72 ± 0.02 ^b^	0.56 ± 0.01 ^cd^	6318.53 ± 18.34 ^d^	4951.69 ± 139.16 ^b^	0.47 ± 0.01 ^a^
N6	9361.89 ± 374.31 ^c^	−123.24 ± 2.16 ^c^	0.76 ± 0.01 ^a^	0.53 ± 0.02 ^cd^	6236.82 ± 97.93 ^d^	3137.12 ± 207.29 ^c^	0.48 ± 0.01 ^a^
N7	8731.18 ± 82.69 ^c^	−143.79 ± 5.05 ^d^	0.70 ± 0.01 ^b^	0.49 ± 0.04 ^e^	4229.80 ± 21.19 ^e^	2495.80 ± 82.20 ^d^	0.22 ± 0.01 ^d^

Data represent the mean of three independent experiments ± standard deviation (*n* = 3). Values with different superscript letters in the same column (lowercase letters) differ significantly (*p* < 0.05). N1–N7 respectively represent noodles with 0%, 10%, 20%, 30%, 40%, 50%, and 60% HTBF.

## Data Availability

Data is contained within the article.

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
