# Peer review of "Effects of Heat-Moisture Treatment Whole Tartary Buckwheat Flour on Processing Characteristics, Organoleptic Quality, and Flavor of Noodles"

_foods, 2022, doi:10.3390/foods11233822_

Round 1

Reviewer 1 Report

This manuscript presents interesting data that both noodle quality and functionality can be improved by using heat-moisture treatment tartary buckwheat flour. The information in the study should be of interest to those who are concerned with production and processing of tartary buckwheat. However, I have some comments that the authors might consider. 

Comment 1

According to the description written in lines 106-108, the amount of soybean protein and wheat gluten which are used for making noodle is not clear. Please add description about the amount in detail. 

Comment 2

In Figure 3, each bar is too thin, and it is somewhat difficult to distinguish each bar. I recommend changing the size of the graph. 

Comment 3

The rule of the superscript letters (a, b, c, d, etc.) showing significant difference in a Table is not consistent. For example, in Table 1, “a” is marked with the lowest value of pasting temperature (79.90) and peak time (5.67). However, “a” is marked with the highest value of peak viscosity (2290.00), trough viscosity (1859.00), breakdown (431.00), final viscosity (2997.00), and setback (1138.00). I think that the letter showing significant difference should be marked from highest values to lowest values in alphabetical order. I recommend unifying the rule of alphabetical order.

Reviewer 2 Report

The manuscript “Effects of heat-moisture treatment whole tartary buckwheat flour on processing characteristics, organoleptic quality and flavor of noodles” deals with effect of various heat treatments on tartary buckwheat flour on various physicochemical properties and organoleptic quality of noodles prepared from it. The manuscript is well-designed and executed. The parameters and content in the manuscript is limited but to the point. The authors developed noodles that heat moisture treatment for making noodles high in resistant starch content. The manuscript needs to be revised accordingly before the final decision is made.

Comments

1.     LN 25-26: Please rewrite the line.

2.     LN 26: Please check the word after full stop.

3.     LN 38: Kindly cite some latest reference.

4.     LN 37: I could not understand the meaning of the line. Kindly check.

5.     LN 72: Kindly mention few countries with proper reference.

6.     Ln 191-194: These lines could be shifted to the end of the paragraph. It seems this line is the output of the current study.

7.     From my point of view, the line graph in Fig. 1 is not valid. Line graph is generally provided where the time or chronological order is maintained. Merely, providing line graphs in various treatment might not be valid. The authors can defend my comment accordingly. Similarly for Fig. 2.

8.     Fig. 3 is fine in all respect. However, the author can provide the error bar.

9.     The roles of volatile alkanes in the food component, particularly in noodles, can be discussed in a few lines in the introduction section.

10.  Kindly revise the English grammar throughout the manuscript.

11.  The formatting and punctuation need to be checked carefully throughout the manuscript

12.  The conclusion is too short. They must add more facts in the conclusion part, along with the addition of the future impact of the present study. 

Reviewer 3 Report

I would add the glycemic index to the keywords.

I haven't seen such a well-written introduction to work for a long time. The authors discuss the nutritional properties of the raw material. They indicate its limitations as to the goal assumed in the work. Then they discuss the property modification technique and the benefits it can bring with great ignorance.

WAI index - equation 1, in my opinion it should also be x100

Methodology - the range of experiments described is sufficient to achieve the goal of the work. Each experience is described so thoroughly that it allows the potential reader to repeat it. The discussion and presentation of the results is prepared at a high level. Clearly legible presentation of the results and their discussion with the available literature. Conclusions properly written, as they should be GENERAL .. but I miss an indication of what parameters should be given to make the pasta tasty, soft and with a low glycemic index - optimal in these conditions.

Author Response

Please check the attachment, thanks.

Round 2

Reviewer 1 Report

The authors have responded appropriately to my comments.
I have no further comments.

Reviewer 2 Report

I congratulate authors for good work.